# Mitigating Hallucinations via Inter-Layer Consistency Aggregation in Large Vision-Language Models

## Abstract

Despite the impressive capabilities of Large Vision-Language Models (LVLMs), they remain susceptible to hallucinations—generating content that is inconsistent with the input image. Existing training-free hallucination mitigation methods often suffer from unstable performance and high sensitivity to hyperparameter settings, limiting their practicality and broader adoption. In this paper, we propose a novel decoding mechanism, ***Decoding with Inter-layer Consistency via Layer Aggregation*** (DCLA), which requires no retraining, fine-tuning, or access to external knowledge bases. Specifically, our approach constructs a dynamic semantic reference by aggregating representations from previous layers, and corrects semantically deviated layers to enforce inter-layer consistency. The method allows DCLA to robustly mitigate hallucinations across multiple LVLMs. Experiments on hallucination benchmarks such as MME and POPE demonstrate that DCLA effectively reduces hallucinations while enhancing the reliability and performance of LVLMs.

## 1 Introduction

Large Vision-Language Models (LVLMs) have rapidly advanced in recent years, demonstrating impressive capabilities in aligning visual and textual modalities, which has notably enhanced their performance on multi-modal tasks such as visual question answering (VQA) and image captioning Bai et al. (2023); Dai et al. (2023); Liu et al. (2023b); Ye et al. (2024); Zhou et al. (2023a); Zhu et al. (2023). Despite these advancements, LVLMs remain susceptible to hallucinations—generating syntactically plausible but visually ungrounded outputs Liu et al. (2023a); Gunjal et al. (2024); Li et al. (2023); Lovenia et al. (2023). This issue severely compromises their reliability and limits their applicability in high-stakes fields such as medical report generation Hartsock & Rasool (2024), autonomous driving Zhou et al. (2024a), and embodied AI systems Ma et al. (2024), where the accuracy and trustworthiness of generated text are crucial.

Recent studies have identified several causes of hallucinations in LVLMs, including over-reliance on statistical biases in training data such as object co-occurrence and background context Li et al. (2023); Chen et al. (2024); Zhou et al. (2023b), the dominance of language priors over visual inputs during decoding Guan et al. (2024); Han et al. (2022); Kaul et al. (2024), and weak cross-modal attention in deeper layers that undermines visual-textual alignment An et al. (2024); Yang et al.. To address hallucinations in LVLMs, knowledge editing methods have been proposed Jiang et al. (2024); Chen et al. (2025); Khandelwal et al. (2024); Zhou et al. (2023b); Perry et al. (2025). These approaches typically aim to mitigate hallucinations by fine-tuning specific memory-related parameters within LVLMs Jiang et al. (2024); Chen et al. (2025); Khandelwal et al. (2024) or by injecting and revising factual information through external knowledge bases Zhou et al. (2023b); Perry et al. (2025). However, such methods generally treat hallucination as a static knowledge deficiency, overlooking the fact that information representations evolve dynamically across layers during inference.

Recently, several studies Chuang et al. (2023); Wang et al.; Leng et al. (2024) have approached this problem from a training-free perspective. Wang et al. observed that hallucinations in LVLMs tend to manifest as localized surges at the later layers, suppressing pre-existing and visually grounded information in the decoding distribution. Based on this observation, they proposed a training-free

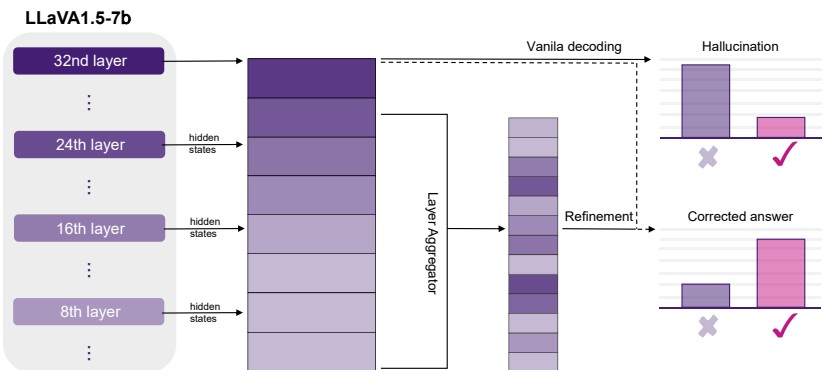

Figure 1: Illustration of Decoding with Inter-Layer Consistency via Layer Aggregation (DCLA) in LLaVA1.5-7b. Vanilla decoding process produces hallucinations and thus generates incorrect answers. By aggregating hidden states across layers to refine the representation, DCLA effectively suppresses hallucinations and restores the ground-truth answer.

approach that mitigates hallucinations by injecting accumulated momentum into the information flow during inference. This design effectively suppresses the localized surges in the decoding distributions observed in later layers. However, their approach focused primarily on accumulating momentum across layers to guide the directional update of activations in later layers, without explicitly addressing the evolving semantic inconsistencies that occur across layers during inference. Consequently, factual information captured by earlier layers may still be attenuated or overridden by semantically divergent activations in later layers, ultimately leading to hallucinations. Additionally, their method is highly sensitive to hyperparameters.

To address these challenges, we propose a novel decoding mechanism, **D**ecoding with Inter-layer **C**onsistency via **L**ayer **A**ggregation (DCLA), shown in Figure 1. Unlike the momentum-based update in DAMO Wang et al., we build an explicit semantic reference across different layers. This resilient reference enables the model to retrieve the correct information captured by earlier layers when decoding in later layers, effectively suppressing the localized surges in the decoding distribution. Specifically, DCLA constructs this inter-layer semantic reference by performing weighted aggregation over the representations of all preceding layers, thereby preserving factual information captured by earlier layers and enhancing cross-layer consistency during decoding.

Experiments on the MME and POPE benchmarks demonstrate that, without any additional training, DCLA significantly reduces hallucinations across four diverse LVLMs: LLaVA1.5-7b Liu et al. (2023a), LLaVA1.5-13b Liu et al. (2023a), LLaVA-NEXT Liu et al. (2024), and mPLUG-Owl2 Ye et al. (2023). Results on VizWiz and MM-Vet datasets show the broader applicability of our method beyond hallucination mitigation. The contributions are summarized as follows:

- We propose Decoding with Inter-layer Consistency via Layer Aggregation (DCLA), a training-free decoding method that reduces hallucinations by enforcing semantic consistency across transformer layers.

- We conduct extensive experiments to validate the proposed DCLA approach, demonstrating that enforcing inter-layer consistency during inference can effectively reduce hallucinations in LVLMs across multiple benchmarks and models.

## 2    RELATED WORK

**Layer Aggregation Mechanisms**    In the field of computer vision, Donahue et al. (2014); Yosinski et al. (2014) have pointed out that as the depth of neural networks increases, high-level representations gain stronger semantic abstraction capabilities, while fine-grained features such as edges and textures are often gradually forgotten. To address this, multi-level feature aggregation mechanisms have been proposed to enhance semantic representation and structural perception Yu et al. (2018). Building on this, the concept of layer aggregation has been widely adopted for efficient feature integration

and mitigating the loss of shallow features Huang et al. (2020); Zhao et al. (2021). In the domain of large foundation models, the idea of layer aggregation has also been extensively adopted Tenney et al. (2019); Brandon et al. (2024); Wu & Tu (2024); Zhou et al. (2024b); Li et al. (2025b). To alleviate memory consumption and improve throughput during inference, Brandon et al. (2024); Wu & Tu (2024); Zhou et al. (2024b) have adopted layer aggregation strategies to reduce the size of key value caches, effectively accelerating model execution. Meanwhile, other studies Tenney et al. (2019); Li et al. (2025b) have explored the use of layer aggregation in the training or fine-tuning stages, demonstrating its effectiveness in enhancing the representation capacity and task-specific performance.

**Hallucinations in LVLMs**  Before large language models (LLMs) emerged, the NLP community primarily defined hallucination as the generation of illogical or source-inconsistent content Lin et al. (2021); Ji et al. (2023); Shi et al. (2023); Agarwal et al. (2018). With the advent of LLMs, hallucinations have become a widely studied phenomenon, particularly in contexts where the generated text deviates from the input or factual reality Zhu et al. (2024); Yao et al. (2023); Xu et al. (2024). In LVLMs, visual and textual information must remain tightly aligned, making hallucinations harder to mitigate and increasing the probability of their occurrence Li et al. (2023); Liu et al. (2023a); Lovenia et al. (2023), especially in tasks such as image captioning and visual question answering Yin et al. (2024); Li et al. (2025a).

Some researchers have actively worked on constructing more refined datasets to fine-tune existing LVLMs Wang et al. (2024); Xiao et al. (2025) or to train correction modules that detect and reconstruct outputs with fewer hallucinations Gunjal et al. (2024); Liu et al. (2023a). However, these approaches often rely on acquiring additional datasets, performing fine-grained tuning on pretrained models, or leveraging external pretrained models. These steps tend to be time-intensive, resource-demanding, and costly in terms of computation. In addition, other studies Kim et al. (2023); Zeng et al. (2021) have focused on mitigating hallucinations by enhancing the alignment between visual and textual modalities.

In recent years, DoLa Chuang et al. (2023) has been proposed to mitigate hallucinations in LLMs through contrastive decoding, achieving promising results without requiring additional training. This idea was later extended to the domain of LVLMs to address hallucinations in multimodal settings. For example, VCD Leng et al. (2024) is a training-free method that compares the output distributions generated from original and distorted visual inputs, thereby enhancing image-relevant information and alleviating object hallucinations. To reduce hallucinations within the hidden state hierarchy during inference, DAMO Wang et al. introduced a momentum mechanism by accumulating activations layer by layer to correct the hidden states of later layers, thus alleviating the concentrated emergence of hallucinations in deeper layers.

## 3  METHOD

### 3.1  DECODING IN LARGE VISION-LANGUAGE MODELS

Large Vision-Language Models (LVLMs) generate textual output in an autoregressive manner through $N$ stacked transformer layers Vaswani et al. (2017); Liu et al. (2023b); Zhu et al. (2023), where each layer plays a crucial role in the inference process. As information propagates forward through the network, feature representations are gradually transformed from low-level signals to high-level semantic representations Rogers et al. (2021). At time step $t$, given an initial fused representation $x_t$, the forward process can be formulated as follows:

$$\mathbf{h}_t^{(i)} = \begin{cases} \text{Embedding}(x_t), & \text{if } i = 0 \\ \text{TransformerLayer}^{(i)}(\mathbf{h}_t^{(i-1)}), & \text{if } i = 1, \dots, N \end{cases} \quad (1)$$

Where $h_t^i$ denotes the hidden states at layer $i$ and time step $t$, and TransformerLayer$^{(i)}$ represents the $i$-th transformer block, consisting of a multi-head self-attention mechanism and a feedforward neural network. The token prediction process, including both standard decoding from the final layer and optional early exit decoding from intermediate layers, can be generalized as follow:

$$p_i(x_{t+1} \mid x_{:t}) = \text{softmax}(\phi(h_t^i)), \quad i = 0, ..., N \quad (2)$$

where $\phi(\cdot)$ is the language modeling head that maps hidden states to vocabulary distributions. This unified formulation enables predictions from any layer, providing flexibility for efficient inference and layer-wise interpretability. Notably, the case $i = N$ corresponds to standard decoding using the final-layer hidden state, while $i < N$ indicates early exit from an intermediate layer.

## 3.2 MOTIVATION

Recent work Wang et al. utilized the early exit mechanism to trace the evolution of token probabilities across the multi-layer inference process in LVLMs. Their findings indicate that hallucinations often manifest as localized surges at the later layers, which tend to override earlier, visually grounded information. To address this, they introduced a momentum-based correction method that aggregates activations across layers and utilizes the accumulated momentum to adjust the hidden states in the later layers.

While effective to some extent, this correction mechanism is highly sensitive to hyperparameters and does not directly address the fundamental issue of inconsistency between layers. We argue that this inter-layer inconsistency is a critical factor undermining semantic stability during decoding.

Motivated by this, we propose the **DCLA** mechanism, which enhances inter-layer consistency by aggregating earlier-layer representations to stabilize semantics throughout the decoding process. This approach effectively mitigates hallucinations caused by semantic drift, improving both the factual accuracy and the robustness of the generated output.

## 3.3 DECODING WITH INTER-LAYER CONSISTENCY VIA LAYER AGGREGATION

**Layer Aggregation**  In order to enforce inter-layer consistency, it is crucial to establish a stable semantic reference throughout the decoding process. Earlier layers encode basic semantic structures that remain robust to overfitting and noise introduced in deeper layers. Therefore, the use of historical representation can effectively anchor the inference trajectory of models, promote consistency across layers, and mitigate hallucinations. We perform weighted aggregation over all earlier layers to ensure that the semantic reference comprehensively integrates the complete semantic information. Formally, at the $i$-th layer, the aggregated representation is defined as:

$$H_{\text{agg}}^{(i)} = \sum_{j=0}^{i-1} \tilde{w}_j^{(i)} \cdot \tilde{h}_j, \quad \text{where} \quad \tilde{h}_j = \begin{cases} \hat{h}_j, & \text{if } j \in \mathcal{C} \\ h_j, & \text{otherwise} \end{cases} \tag{3}$$

Here, $\tilde{h}_j$ denotes the effective hidden state of the $j$-th layer, which dynamically incorporates correction information. If layer $j$ has undergone a refinement process, the corrected hidden state $\hat{h}_j$ is used. Otherwise, the original hidden state $h_j$ is retained. The set $\mathcal{C} \subseteq \{0, 1, \ldots, i-1\}$ records all layers that have undergone correction prior to the $i$-th layer. Therefore, the mechanism selectively integrates the most reliable semantic information available at each layer. This design ensures that once a layer undergoes refinement, the corrected hidden state persistently contributes to subsequent decoding, effectively mitigating hallucinations. To prioritize layers that are closer to the current decoding step while still leveraging the semantic stability provided by earlier layers, we introduce a normalized exponential weighting scheme. This design reflects the intuition that recent layers contain more task-specific contextual refinements, whereas earlier layers offer foundational but potentially less context-aware semantics. The weight assigned to each layer $j$ is defined as:

$$\tilde{w}_j^{(i)} = \frac{\exp\left(s(j, i)\right)}{\sum\limits_{k=0}^{i-1} \exp\left(s(k, i)\right)} \tag{4}$$

where $\gamma$ controls the decay based on layer distance. $s(j, i) = j - (i - 1)$ is a distance function that captures the relative positional relationship between layer $j$ and layer $i$. Compared to linear or uniform weighting strategies, this approach provides a more flexible and natural decay pattern, which aligns with the hierarchical nature of transformer representations where semantic granularity deepens progressively across layers.

**Decoding with Inter-Layer Consistency**  In standard LVLMs, the decoding process typically follows the vanilla decoding strategy, where each transformer layer updates its hidden state solely

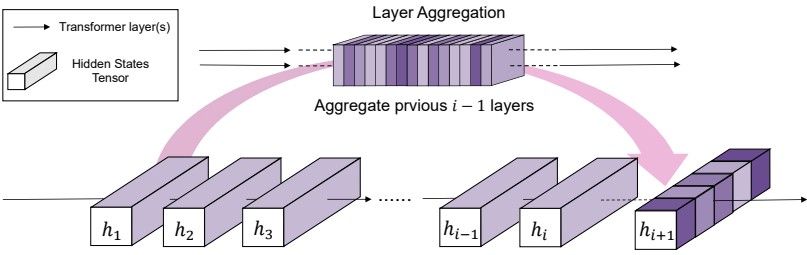

Figure 2: Layer Aggregation mechanism to form a stable semantic reference.

based on the output of the preceding layer. However, this unidirectional information flow can lead to the gradual loss of fine-grained semantic cues captured by earlier layers. To address this issue, we integrate the aggregated representation $H_{\text{agg}}^{(i)}$, into the decoding process. The hidden state is corrected through a linear fusion mechanism.

$$\hat{h}_i = \alpha \cdot h_i + (1 - \alpha) \cdot H_{\text{agg}}^{(i)} \tag{5}$$

where $\alpha$ controls the balance between the original representation and the aggregated semantic reference. As shown in Figure 2, this correction mechanism allows the current layer to benefit from both localized contextual information and global aggregated semantics.

### 3.4 ADAPTIVE LAYER AGGREGATION CORRECTION

Tenney et al. (2019) have shown that different layers of the model perform different roles in semantic construction. Earlier layers focus more on capturing lower-level information, while later layers progressively inject abstract semantics and factual knowledge. Therefore, applying indiscriminate corrections to all layers may disrupt the natural evolution of information within the model.

To verify this hypothesis, we conducted preliminary experiments on MME dataset, and the results in Table 1 revealed that correcting different layers led to significant fluctuations in overall model performance. Moreover, increasing the number of corrected layers did not necessarily result in better outcomes. These observations suggest that correction should not be applied indiscriminately but rather selectively based on the intrinsic characteristics of each layer.

Table 1: Perception performance of LLaVA1.5-7b on the MME dataset with different correction ranges. Each column indicates a hallucination category, and **bold** values denote the best results.

| Correction | Exist. | Count | Pos. | Color | Posters | Celeb. | Scene | Land. | Art. | OCR | Total |
|---|---|---|---|---|---|---|---|---|---|---|---|
| baseline | 190.00 | **160.00** | 138.33 | 165.00 | 140.48 | 135.00 | 156.25 | 162.25 | 119.25 | 125.00 | 1491.56 |
| 1–16 layers | **190.00** | 145.00 | **138.33** | **180.00** | 141.16 | **132.94** | **161.75** | 165.50 | **122.25** | 140.00 | **1516.93** |
| 1–20 layers | 190.00 | 145.00 | 138.33 | 175.00 | 141.16 | 132.06 | 160.25 | 167.00 | 119.25 | **147.50** | 1515.55 |
| 1–24 layers | 190.00 | 150.00 | 133.33 | 170.00 | 142.18 | 132.94 | 161.00 | 167.00 | 120.00 | 145.00 | 1511.45 |
| 1–28 layers | 190.00 | 140.00 | 138.33 | 180.00 | 141.16 | 135.88 | 158.75 | 163.75 | 119.50 | 147.50 | 1514.87 |
| 1–32 layers | 185.00 | 120.00 | 108.33 | 160.00 | **155.78** | 135.88 | 160.00 | 161.75 | 109.75 | 107.50 | 1404.00 |

Thus, we propose a dynamic layer correction mechanism. During inference, the model adaptively determines whether a hallucination surge has occurred at each layer by comparing the semantic features of the current hidden states with the global aggregated vector. The dynamic layer selection mechanism, together with our layer aggregation strategy, completes our overall approach, Decoding with Inter-layer Consistency via Layer Aggregation (DCLA). To more directly capture the intrinsic semantic changes, we focus on the hidden states themselves rather than their derived probability distributions. Specifically, we do not employ an early exit mechanism to obtain intermediate output distributions. Instead, we flatten the hidden states at the $i$-th layer to obtain $h_i^{\text{flat}}$, and compute the cosine similarity with $H_{\text{agg}}^{(i),\text{flat}}$.

$$\text{cos\_sim}(h_i, H_{\text{agg}}^{(i)}) = \frac{h_i^{\text{flat}} \cdot H_{\text{agg}}^{(i),\text{flat}}}{\|h_i^{\text{flat}}\| \|H_{\text{agg}}^{(i),\text{flat}}\|} \tag{6}$$

Specifically, if the cosine similarity falls below a predefined threshold $\tau$, we regard the current layer as exhibiting unstable representation dynamics and activate a correction mechanism to refine the hidden states. Otherwise, if the similarity exceeds $\tau$, the decoding proceeds without modification. This adaptive strategy ensures that corrections are only applied when necessary, preserving the model's natural semantic progression while mitigating potential inconsistencies.

## 4 EXPERIMENT

### 4.1 SETUP

**Datasets**  To comprehensively validate the effectiveness of our DCLA method in mitigating hallucination issues in Large Vision-Language Models (LVLMs), we utilize the MME benchmark Fu et al. (2023). This benchmark comprises 14 diverse tasks, which are categorized under perception and cognition. Additionally, we specifically investigate the effectiveness of DCLA in addressing object hallucinations through benchmark POPE (Polling-based Object Probing Evaluation) Li et al. (2023), which uses the SEEM-annotated datasets MSCOCO Lin et al. (2014), A-OKVQA Schwenk et al. (2022) and GQA Hudson & Manning (2019). In addition, we further assess the generalization of our method on two real-world VQA benchmarks: VizWiz Gurari et al. (2018), which features noisy and ambiguous image inputs, and MM-Vet Yu et al. (2023), which provides a comprehensive assessment of multimodal models across six core capabilities. These datasets provide a practical testbed for evaluating the robustness of LVLMs in open-domain settings.

**Models and Baselines**  We conduct experiments on four recent LVLMs to validate the generalization ability of our method. These include two models with identical architecture but different parameter scales: LLaVA1.5-7b and LLaVA1.5-13b Liu et al. (2023b), as well as two 7b-scale models with distinct vision-language fusion and pre-training strategies: LLaVA-NEXT Liu et al. (2024) and mPLUG-Owl2 Ye et al. (2023). For baseline comparisons, we evaluate DCLA against several representative decoding methods such as regular decoding, VCD Leng et al. (2024), DoLa Chuang et al. (2023), and DAMO Wang et al.. To ensure the fairness and reproducibility of our comparisons, all decoding strategies are evaluated under decoding temperature consistently set to zero throughout all experiments.

**Hyperparameters Setting**  We use only two hyperparameters in our experiments, namely $\tau$ and $\alpha$, with their specific values provided in the Appendix. To further verify the effectiveness of our hyperparameter choices and the contribution of each component, we conduct an ablation study to compare different combinations. The detailed results are also included in the Appendix. Experimental results demonstrate that our chosen hyperparameter configuration maintains stable and strong performance across different settings.

Table 2: Experimental results of various decoding strategies on MME dataset across four models: LLaVA1.5-7b, LLaVA-NEXT, LLaVA1.5-13b, and mPLUG-Owl2.

| Model | Decoding | Existence | Count | Position | Color | Posters | Celebrity | Scene | Landmark | Artwork | OCR | Total |
|-------|----------|-----------|-------|----------|-------|---------|-----------|-------|----------|---------|-----|-------|
| LLaVA1.5-7b | Regular | 190.00 | 160.00 | 138.33 | 165.00 | 140.48 | 135.00 | 156.25 | 162.25 | 119.25 | 125.00 | 1491.56 |
| | VCD | 190.00 | 163.33 | 133.33 | 158.33 | 129.59 | **139.12** | 155.75 | **166.50** | **124.00** | 125.00 | 1484.96 |
| | DoLa | 190.00 | 153.33 | 143.33 | 165.00 | 141.50 | 132.35 | 157.75 | 160.50 | 118.75 | 132.50 | 1495.02 |
| | DAMO | **195.00** | 150.00 | 143.33 | 165.00 | **144.56** | 134.12 | **157.75** | 163.75 | 120.00 | 140.00 | 1513.51 |
| | **DCLA** | 190.00 | 163.33 | 148.33 | 175.00 | 137.41 | 132.06 | 156.25 | 160.50 | 117.25 | **140.00** | **1520.14** |
| LLaVA-NEXT | Regular | 195.00 | 135.00 | 143.33 | 170.00 | 159.52 | 142.94 | **162.25** | 155.75 | 123.00 | 132.50 | 1519.30 |
| | VCD | 175.00 | 125.00 | 95.00 | 140.00 | 148.98 | 145.29 | 159.00 | **169.75** | 130.25 | 130.00 | 1418.27 |
| | DoLa | 190.00 | 133.33 | 143.33 | 170.00 | 132.65 | **155.59** | 156.25 | 135.00 | **136.75** | 162.50 | 1515.41 |
| | DAMO | 195.00 | 130.00 | 133.33 | 160.00 | 149.32 | 145.29 | 159.50 | 143.25 | 123.75 | 132.50 | 1471.95 |
| | **DCLA** | **195.00** | 140.00 | 143.33 | 170.00 | **160.20** | 142.94 | 161.50 | 155.75 | 124.50 | 132.50 | **1525.73** |
| LLaVA1.5-13b | Regular | 188.33 | 145.00 | 123.33 | 160.00 | 159.52 | 159.71 | 157.25 | 141.75 | 121.75 | 147.50 | 1504.15 |
| | VCD | 190.00 | **163.33** | 120.00 | **175.00** | 151.70 | 159.41 | 158.25 | 129.00 | **125.25** | 132.50 | 1504.44 |
| | DoLa | 190.00 | 150.00 | 123.33 | 160.00 | 160.54 | 157.06 | 155.75 | 134.50 | 124.00 | 147.50 | 1502.69 |
| | DAMO | **190.00** | 125.00 | 113.33 | 150.00 | **166.66** | 152.06 | **163.25** | **166.50** | 107.75 | 140.00 | 1474.56 |
| | **DCLA** | 188.33 | 145.00 | 123.33 | 160.00 | 159.52 | **160.88** | 158.00 | 140.50 | 121.75 | **147.50** | 1504.82 |
| mPLUG-Owl2 | Regular | 185.00 | 165.00 | 78.33 | 150.00 | **163.27** | 162.94 | 152.75 | 160.00 | 139.75 | 102.50 | 1459.54 |
| | VCD | 180.00 | 160.00 | 61.67 | 151.67 | 151.36 | 108.82 | 158.00 | 115.00 | 130.00 | 95.00 | 1311.52 |
| | DoLa | 175.00 | 160.00 | **93.33** | **163.33** | 155.78 | 160.88 | **159.25** | 142.25 | **142.25** | **110.00** | 1462.33 |
| | DAMO | 180.00 | 155.00 | 78.33 | 145.00 | 134.01 | **167.35** | 154.00 | **168.25** | 133.50 | 87.50 | 1402.95 |
| | **DCLA** | **185.00** | **165.00** | 78.33 | 155.00 | 162.24 | 163.82 | 153.00 | 159.50 | 139.00 | 102.50 | **1463.40** |

## 4.2 RESULTS

**Results on MME**  To systematically assess the effectiveness of DCLA in mitigating hallucinations, we adopt the perception subset of the MME benchmark, which comprises 10 tasks and has been widely used in recent studies related to hallucinations. Our experiments are conducted on some representative LVLMs. As shown in Table 2, DCLA consistently outperforms almost all baseline decoding methods across the board, achieving total scores of 1520.14 on LLaVA1.5-7b and 1525.73 on LLaVA-NEXT. Notably, DCLA achieves a total score of 1504.82 on the larger-scale model LLaVA1.5-13b. Furthermore, it maintains strong performance on a structurally different architecture, mPLUG-Owl2, with a total score of 1463.40, which surpasses other decoding-based baselines such as DoLa and DAMO. Existing decoding-based approaches do not guarantee consistent improvements across models: DoLa has no significant improvement over LLaVA1.5-7b, VCD struggles to maintain stable performance across almost all architectures, and DAMO results in performance degradation on LLaVA-NEXT. These findings highlight the superior generation ability and reliability of DCLA in hallucination mitigation for LVLMs.

Table 3: Experimental results of various decoding strategies on the Random and Popular subsets of the SEEM-annotated MSCOCO, A-OKVQA, and GQA datasets from POPE using four models: LLaVA1.5-7b, LLaVA-NEXT, LLaVA1.5-13b, and mPLUG-Owl2.

| Setting | Model | Decoding | MSCOCO | | A-OKVQA | | GQA | |
|---|---|---|---|---|---|---|---|---|
| | | | Accuracy | F1 Score | Accuracy | F1 Score | Accuracy | F1 Score |
| Random | LLaVA1.5-7b | Regular | 89.60 | 89.72 | 87.23 | 88.44 | 86.87 | 88.14 |
| | | VCD | 89.07 | 89.00 | 86.37 | 87.44 | 86.00 | 87.33 |
| | | DoLa | 89.70 | 89.79 | 86.10 | 87.48 | 85.47 | 87.00 |
| | | DAMO | 89.94 | 89.90 | 87.83 | 88.88 | 86.03 | 87.52 |
| | | **DCLA** | **90.03** | **89.99** | **87.93** | **88.98** | **87.90** | **88.94** |
| | LLaVA-NEXT | Regular | 88.83 | 87.58 | 91.07 | 90.87 | 90.03 | 89.34 |
| | | VCD | 80.00 | 75.14 | 81.13 | 77.45 | 81.13 | 77.29 |
| | | DoLa | 85.40 | 83.02 | 88.73 | 87.68 | 86.97 | 85.34 |
| | | DAMO | **88.90** | **87.69** | 90.87 | 90.43 | 88.63 | 87.86 |
| | | **DCLA** | 88.87 | 87.60 | **91.10** | **90.69** | **90.10** | **89.52** |
| | LLaVA1.5-13b | Regular | 88.37 | 87.15 | 91.03 | 90.72 | 91.03 | 90.75 |
| | | VCD | 87.10 | 85.50 | 89.00 | 88.33 | 89.43 | 88.90 |
| | | DoLa | 88.30 | 87.04 | 90.77 | 90.37 | 91.03 | 90.67 |
| | | DAMO | 90.03 | 89.59 | **91.60** | **91.78** | 90.90 | **91.09** |
| | | **DCLA** | **91.07** | **90.83** | 91.50 | 91.28 | 91.00 | 90.70 |
| | mPLUG-Owl2 | Regular | 88.40 | 87.71 | 88.09 | 88.17 | 86.10 | 85.41 |
| | | VCD | 82.17 | 79.96 | 82.83 | 81.49 | 81.73 | 80.25 |
| | | DoLa | 86.97 | 85.60 | 87.63 | 86.96 | 84.77 | 83.06 |
| | | DAMO | **88.63** | **88.19** | **88.26** | **88.43** | 86.70 | 86.23 |
| | | **DCLA** | 88.53 | 87.99 | 88.10 | 88.16 | **86.80** | **86.25** |
| Popular | LLaVA1.5-7b | Regular | 86.20 | 86.81 | 80.10 | 83.07 | 74.50 | 79.29 |
| | | VCD | 85.63 | 86.03 | 78.90 | 81.82 | 73.73 | 78.60 |
| | | DoLa | 86.07 | 86.67 | 80.40 | 83.21 | 75.30 | 79.75 |
| | | DAMO | 86.67 | 87.06 | 81.07 | 83.70 | **76.17** | **80.29** |
| | | **DCLA** | **86.73** | **87.10** | **81.13** | **83.78** | 75.20 | 79.68 |
| | LLaVA-NEXT | Regular | 87.63 | 86.43 | 89.13 | 88.87 | 86.57 | 86.23 |
| | | VCD | 79.83 | 74.99 | 80.90 | 77.23 | 79.20 | 75.53 |
| | | DoLa | 85.03 | 82.67 | 87.83 | 86.83 | 85.07 | 83.55 |
| | | DAMO | 87.70 | **86.54** | 89.03 | 88.72 | 85.97 | 85.43 |
| | | **DCLA** | **87.73** | 86.51 | **89.23** | **88.96** | **86.57** | **86.30** |
| | LLaVA1.5-13b | Regular | 87.53 | 86.36 | 89.13 | 88.97 | 88.43 | 88.38 |
| | | VCD | 86.13 | 84.58 | 86.73 | 86.26 | 86.53 | 86.28 |
| | | DoLa | 87.53 | 86.31 | 88.93 | 88.68 | **88.60** | **88.43** |
| | | DAMO | 88.87 | 88.51 | 89.07 | **89.56** | 86.77 | 87.55 |
| | | **DCLA** | **90.27** | **90.09** | **89.17** | 89.14 | 88.40 | 88.33 |
| | mPLUG-Owl2 | Regular | 86.56 | 86.04 | 84.33 | 84.18 | 79.37 | 80.38 |
| | | VCD | 80.13 | 77.89 | 80.77 | 79.93 | 77.83 | 77.12 |
| | | DoLa | 85.77 | 84.48 | 84.47 | 84.15 | 81.37 | 80.29 |
| | | DAMO | 86.50 | 86.28 | 84.07 | 84.92 | **79.93** | **80.59** |
| | | **DCLA** | **86.83** | **86.45** | **84.43** | **85.06** | 79.87 | 80.44 |

**Results on POPE** To evaluate the effectiveness of our proposed method in mitigating object-level hallucinations, we conduct experiments on the SEEM-annotated versions of the MSCOCO, A-OKVQA, and GQA datasets provided by the POPE benchmark. Each dataset is further divided into three subsets: adversarial, popular, and random. As shown in Table 3, we compare DCLA with several representative decoding strategies and report two key evaluation metrics: Accuracy and F1 Score, to comprehensively assess the consistency of object recognition. The results on the popular and random subsets are presented in Table 3, while the results on the adversarial subset are provided in the Appendix.

*MSCOCO datasets* DCLA reduces hallucinations in all evaluation categories across models, demonstrating stable and robust improvements. Notably, under the most challenging adversarial setting of LLaVA1.5-7b, DCLA achieves a 0.93% increase in accuracy and a 0.5% gain in F1 score. In contrast, DoLa, VCD, and DAMO exhibit higher instability and even performance regression in some cases.

*A-OKVQA datasets* On the A-OKVQA dataset, DCLA outperforms most existing decoding strategies across all evaluation settings, demonstrating stable and comprehensive improvements. In the most challenging adversarial setting, DCLA yields an average accuracy improvement of 1.02% across four different models. In contrast, other methods fail to guarantee consistent improvements and occasionally lead to performance regressions.

*GQA datasets* DCLA is a strong and adaptable decoding strategy on the GQA dataset, delivering consistent improvements across diverse models and settings, including base and advanced architectures. Notably, DCLA achieves the best performance on LLaVA-NEXT, surpassing all other decoding methods. In particular, under the random setting of mPLUG-Owl2, DCLA achieves gains of 0.70% in accuracy and 0.84% in F1 score.

Table 4: Evaluation of DCLA and other decoding methods on LLaVA1.5-7b using VizWiz and MM-Vet.

| Decoding | VizWiz | | | | | MM-Vet | | | | | | |
|---|---|---|---|---|---|---|---|---|---|---|---|---|
| | Number | Yes/No | Unans. | Other | **Overall** | Rec | OCR | Know | Gen | Spat | Math | **Total** |
| Regular | 47.62 | 78.26 | 74.30 | 38.11 | 50.05 | 36.1 | **24.5** | 17.5 | 22.2 | 25.7 | **11.5** | 31.5 |
| VCD | 42.70 | 77.64 | 72.54 | **38.12** | 49.47 | 27.7 | 21.5 | 8.3 | 7.6 | 28.1 | 3.8 | 26.1 |
| DoLa | **53.33** | **80.00** | 69.64 | 37.36 | 48.43 | 37.6 | 21.3 | **21.5** | **24.6** | 25.6 | 7.7 | 31.8 |
| DAMO | 48.10 | 78.88 | 71.78 | 36.63 | 48.41 | 35.3 | 21.6 | 21.1 | 21.6 | 24.9 | 7.7 | 31.3 |
| **DCLA** | 45.71 | 78.65 | **77.19** | 37.78 | **50.62** | **37.7** | 23.9 | 19.9 | 24.5 | **29.3** | 7.7 | **32.1** |

**Results on General-Purpose Multimodal Benchmarks** As shown in Table 4, DCLA achieves strong overall performance on general-purpose multimodal benchmarks. It obtains 50.62% Overall accuracy on VizWiz, with 77.19% in the Unanswerable category. On MM-Vet, it records a Total score of 32.1%, including 37.7% in Recognition and 29.3% in Spatial Reasoning. Compared to other decoding methods, DCLA maintains robust and balanced results across all settings, indicating superior generalization.

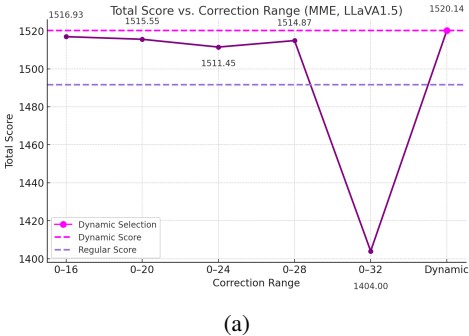

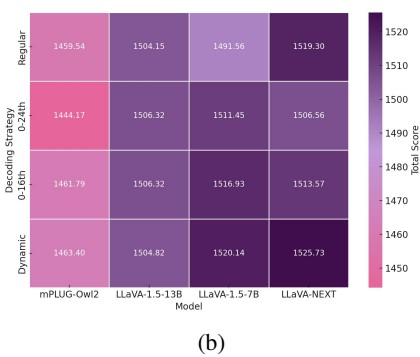

(a)  (b)

Figure 3: (a) Comparison of different fixed refinement layers in DCLA (0–20th, 0–24th, 0–28th, 0–32nd) against standard decoding and our adaptive correction mechanism. (b)Evaluation of the dynamic correction mechanism and fixed correction on LLaVA1.5-7b, LLaVA-Next, LLaVA1.5-13b, and mPLUG-Owl2.

**Evaluation on Adaptive Layer Aggregation Correction** To verify the effectiveness of our dynamic selection mechanism, we conduct a series of ablation studies. In all experiments, aggregation is consistently performed starting from the 1st layer, while representation refinement is applied to each layer from the 0-th up to the $i$-th layer. All other parameters are kept unchanged to ensure a fair and consistent comparison. As shown in Figure 3a, the experimental results on the MME dataset using the LLaVA1.5 model indicate that most fixed refinement layer settings achieve reasonably good performance, except for a few extreme values of $i$. In contrast, the dynamic selection mechanism consistently achieves the highest accuracy across all configurations, demonstrating its flexibility and effectiveness in guiding the decoding process. A similar trend can be observed across the other three models, as illustrated in Figure 3b, further confirming the robustness and generalizability of our approach.

**Effect of Correction Strength and Trigger Threshold** We evaluate the sensitivity of the LLaVA1.5 model to different correction strength values and trigger threshold on the POPE (MSCOCO setting), and record the corresponding accuracy. As shown in Figure 4b, the model exhibits consistent performance fluctuations across the Random, Popular, and Adversarial subsets as $\alpha$ varies from 0.8 to 0.9, and $\tau$ varies from 0.7 to 0.8, indicating the model's sensitivity to this hyperparameter. Notably, when $\alpha = 0.82$ and $\tau = 0.74$, the model achieves the highest accuracy on all three subsets.

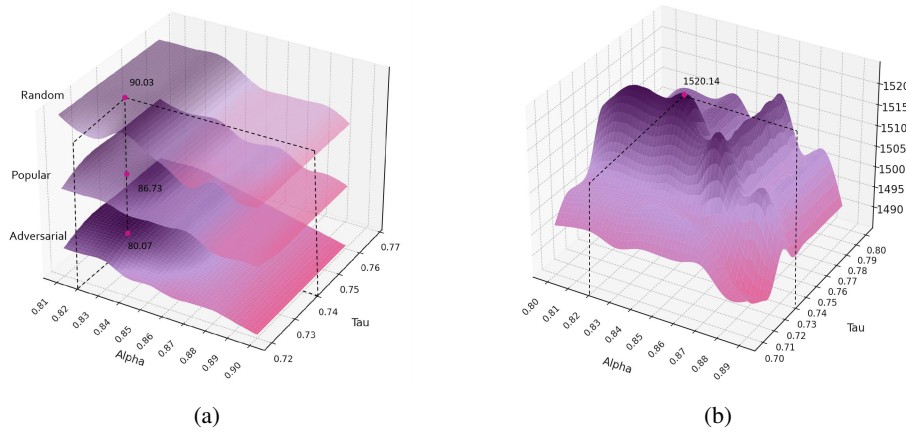

(a)  (b)

Figure 4: (a) Parameter sensitivity analysis of correction strength and trigger threshold on POPE subsets (Random, Popular, Adversarial). (b) Parameter sensitivity analysis of correction strength and trigger threshold on MME dataset.

To further investigate the effectiveness of this parameter setting, we conduct a sensitivity analysis on the MME benchmark. As shown in Figure 4a, LLaVA1.5 is sensitive to the values of both correction strength and trigger threshold. When $\alpha$ and $\tau$ are set to the same values as those used on the POPE, specifically $\alpha = 0.74$ and $\tau = 0.82$, the model achieves the best overall performance on the MME benchmark. This indicates that the selected parameter combination exhibits good cross-dataset generalization on LLaVA1.5.

## 5 CONCLUSION

In this work, we propose Decoding with Inter-layer Consistency via Layer Aggregation (DCLA), a training-free decoding strategy designed to mitigate hallucinations in Large Vision-Language Models (LVLMs). DCLA introduces an explicit cross-layer semantic reference during decoding by aggregating intermediate representations and dynamically selecting and refining the processing layer to enhance semantic stability and suppress hallucinations. Experimental results demonstrate that DCLA achieves consistent improvements on several challenging hallucination evaluation benchmarks, including MME and POPE, with particularly strong gains under adversarial settings. Moreover, DCLA achieves robust improvements on diverse real-world datasets such as VizWiz and MM-Vet, indicating its ability to generalize beyond hallucination mitigation. Notably, DCLA is highly compatible with existing model architectures and can be seamlessly integrated into mainstream multimodal frameworks.

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

# APPENDIX

## A    RESULTS ON ADVERSARIAL SUBSETS IN POPE

Table 5: Experimental results of various decoding strategies on the Adversarial subsets of the SEEM-annotated MSCOCO, A-OKVQA, and GQA datasets from POPE using four models: LLaVA1.5-7b, LLaVA-NEXT, LLaVA1.5-13b, and mPLUG-Owl2.

| Setting | Model | Decoding | MSCOCO | | A-OKVQA | | GQA | |
|---|---|---|---|---|---|---|---|---|
| | | | Accuracy | F1 Score | Accuracy | F1 Score | Accuracy | F1 Score |
| | LLaVA1.5-7b | Regular | 79.77 | 81.78 | 69.20 | 76.02 | 68.33 | 75.50 |
| | | VCD | 79.27 | 81.01 | 68.30 | 74.97 | 68.10 | 75.15 |
| | | DoLa | 79.47 | 81.52 | 69.60 | 76.16 | 68.90 | 75.77 |
| | | DAMO | 80.50 | 82.14 | **70.64** | **76.85** | **70.07** | **76.43** |
| | | **DCLA** | **80.70** | **82.28** | 70.23 | 76.60 | 69.13 | 75.91 |
| | LLaVA-NEXT | Regular | 86.40 | 85.27 | 82.03 | 82.85 | 82.97 | 83.27 |
| | | VCD | 79.13 | 74.34 | 75.70 | 72.72 | 76.23 | 72.98 |
| | | DoLa | 84.30 | 81.97 | 82.83 | 82.37 | 81.87 | 80.71 |
| | | DAMO | 86.33 | 85.26 | **83.16** | **83.68** | 82.53 | 82.49 |
| Adversarial | | **DCLA** | **86.53** | **85.38** | 82.33 | 83.08 | **83.17** | **83.40** |
| | LLaVA1.5-13b | Regular | 85.63 | 84.60 | 81.93 | 82.91 | 83.73 | **84.40** |
| | | VCD | 84.23 | 82.83 | 80.40 | 80.95 | 82.43 | 82.81 |
| | | DoLa | 85.70 | 84.61 | 82.07 | 82.86 | 82.97 | 81.89 |
| | | DAMO | 85.53 | 85.56 | 81.13 | **83.25** | 81.47 | 83.39 |
| | | **DCLA** | **85.77** | **86.14** | **82.07** | 83.22 | **83.73** | 84.37 |
| | mPLUG-Owl2 | Regular | 84.20 | 83.98 | 76.67 | 78.71 | 78.07 | 79.03 |
| | | VCD | 77.73 | 76.21 | 73.50 | 74.26 | 74.43 | 74.53 |
| | | DoLa | 84.00 | 82.88 | **78.17** | 79.14 | 78.07 | 77.80 |
| | | DAMO | 83.33 | 83.59 | 76.20 | 79.04 | 78.03 | 79.13 |
| | | **DCLA** | **84.20** | **84.14** | 77.07 | **79.44** | **78.17** | 79.13 |

## B    LIMITATION ANALYSIS

DCLA also has limitations. This work focuses on hallucination mitigation in image-text multimodal models and does not extend the proposed mechanism to video-language scenarios, where temporal dynamics and cross-frame consistency present new challenges. In addition, DCLA is an inference-only strategy and does not incorporate supervised signals such as reinforcement learning from human feedback or task-specific fine-tuning, which could further enhance its effectiveness. Finally, the method relies entirely on the model's internal representations and does not utilize external retrieval or grounding modules, limiting its ability to correct misinformation acquired during pretraining.

## C    HYPERPARAMETER SETTING

### C.1    BASELINE SETTING

In the DoLa decoding strategy, we fixed the mature layer to the 32nd layer and adopted a dynamic candidate layer selection mechanism with an adaptive plausibility constraint. For the DAMO method, we kept all settings consistent with its official configuration. For the VCD method, we set the decoding temperature to 0. The setting of our method is shown in Table 6.

### C.2    HYPERPARAMETER SETTING IN DCLA

The code of DCLA is available at `https://anonymous.4open.science/r/DCLA-1028/`. Table6 presents the hyperparameter configurations of DCLA across four different models. The correction strength coefficient $\alpha$ and the triggering threshold $\tau$ are adjusted for each model to ensure optimal performance. Specifically, LLaVA1.5-7b, LLaVA-NEXT, LLaVA1.5-13b, and mPLUG-Owl2 are assigned distinct values of $\alpha$ and $\tau$, reflecting their architectural and scale differences. Importantly,

we do not adopt dataset-specific parameter tuning to deliberately improve performance. Instead, each model is assigned a single parameter setting that consistently improves results across most datasets, demonstrating the strong generalization ability of our method as well as its robustness to hyperparameter sensitivity.

Table 6: Hyperparameters of DCLA for different models.

|          | LLaVA1.5-7b | LLaVA-NEXT | LLaVA1.5-13b | mPLUG-Owl2 |
|----------|-------------|------------|--------------|------------|
| $\alpha$ | 0.82        | 0.96       | 0.86         | 0.90       |
| $\tau$   | 0.74        | 0.93       | 0.76         | 0.95       |

## D ABLATION STUDY

### D.1 EVALUATION THE SENSITIVITY OF CORRECTION STRENGTH

We evaluate the sensitivity of the LLaVA1.5-7b model to different correction strength values on the POPE dataset (MSCOCO setting), and report the corresponding Accuracy scores. As shown in Table 7, the model exhibits consistent performance fluctuations across the Random, Popular, and Adversarial subsets as $\tau$ varies from 0.7 to 0.8. Notably, when $\alpha = 0.82$, the model achieves the highest average score of 85.82.

Table 7: The sensitivity of correction strength on POPE (MSCOCO) dataset using LLaVA1.5-7b model

| $\tau$      | 0.80      | 0.82      | 0.84  | 0.86  | 0.88  | 0.90  |
|-------------|-----------|-----------|-------|-------|-------|-------|
| Random      | 90.03     | **90.03** | 90.00 | 89.92 | 89.80 | 89.71 |
| Popular     | **86.80** | 86.73     | 86.69 | 86.53 | 86.39 | 86.20 |
| Adversarial | 80.60     | **80.70** | 80.60 | 80.42 | 80.26 | 80.03 |
| Average     | 85.81     | **85.82** | 85.76 | 85.62 | 85.48 | 85.31 |

### D.2 EVALUATION THE SENSITIVITY OF TRIGGER THRESHOLD

To further investigate the influence of the trigger threshold, we conduct a sensitivity analysis of $\tau$ on the MME benchmark. As shown in Table 8, LLaVA1.5 is quite sensitive to the value of $\alpha$. When $\alpha$ is set to 0.74, the model achieves the best overall performance on the MME benchmark. This highlights that $\alpha = 0.74$ is the optimal setting under our current configuration.

Table 8: The sensitivity of trigger threshold on MME benchmark using LLaVA1.5-7b

| $\tau$ | Exist. | Count | Pos. | Color | Posters | Celeb. | Scene | Land. | Art. | OCR | Total |
|--------|--------|-------|------|-------|---------|--------|-------|-------|------|-----|-------|
| 0.7  | 190.00 | 158.33 | 143.33 | 165.00 | 138.44 | 135.00 | 157.00 | 159.75 | 118.25 | 132.50 | 1497.60 |
| 0.72 | 190.00 | 158.33 | 148.33 | 165.00 | **138.44** | 135.00 | **157.00** | 160.50 | **118.25** | 132.50 | 1503.35 |
| 0.74 | **190.00** | **163.33** | **148.33** | **175.00** | 137.41 | **132.06** | 156.25 | **160.50** | 117.25 | **140.00** | **1520.14** |
| 0.76 | 190.00 | 163.33 | 148.33 | 175.00 | 136.39 | 132.06 | 156.25 | 160.00 | 117.25 | 132.50 | 1511.12 |
| 0.78 | 190.00 | 163.33 | 148.33 | 175.00 | 136.39 | 132.06 | 156.25 | 160.00 | 117.25 | 132.50 | 1511.12 |
| 0.80 | 190.00 | 163.33 | 148.33 | 175.00 | 136.39 | 132.06 | 156.25 | 160.00 | 117.25 | 132.50 | 1511.12 |

### D.3 VALIDATION OF THE SELECTED PARAMETERS ON MME

To validate the effectiveness of our selected hyperparameters, we conducted a larger-scale grid search experiment. Table 9 reports the total scores of the LLaVA1.5-7b model on the perception subset under different combinations of the correction strength $\alpha$ and trigger threshold $\tau$. The table systematically explores values of $\alpha$ from 0.80 to 0.89 and $\tau$ from 0.70 to 0.79. The results reveal that performance is relatively stable across a wide range of settings, with the highest scores observed around $\alpha = 0.82$

and $\tau = 0.74$, suggesting that these hyperparameter ranges offer optimal balance for hallucination mitigation and perceptual grounding.

Table 9: Total Score across of perceptron different $\tau$ and $\alpha$ combinations on LLaVA1.5-7b.

| $\tau\backslash\alpha$ | 0.80 | 0.81 | 0.82 | 0.83 | 0.84 | 0.85 | 0.86 | 0.87 | 0.88 | 0.89 |
|---|---|---|---|---|---|---|---|---|---|---|
| 0.70 | 1496.7 | 1497.7 | 1497.6 | 1495.9 | 1497.4 | 1497.6 | 1497.8 | 1492.5 | 1497.5 | 1493.3 |
| 0.71 | 1501.7 | 1502.7 | 1502.6 | 1502.6 | 1504.1 | 1504.2 | 1497.8 | 1492.5 | 1492.5 | 1491.6 |
| 0.72 | 1501.7 | 1502.7 | 1503.4 | 1503.4 | 1503.9 | 1502.4 | 1500.9 | 1489.4 | 1489.4 | 1489.2 |
| 0.73 | 1504.4 | 1510.0 | 1509.8 | 1509.4 | 1511.1 | 1510.4 | 1518.4 | 1507.1 | 1508.5 | 1496.8 |
| 0.74 | 1514.8 | 1519.8 | **1520.1** | 1519.2 | 1520.1 | 1515.2 | 1516.2 | 1503.6 | 1505.8 | 1493.4 |
| 0.75 | 1512.3 | 1518.8 | 1519.4 | 1519.2 | 1520.2 | 1507.7 | 1516.2 | 1503.6 | 1505.8 | 1493.4 |
| 0.76 | 1504.0 | 1510.5 | 1511.1 | 1511.0 | 1512.7 | 1507.7 | 1516.2 | 1503.6 | 1505.8 | 1493.4 |
| 0.77 | 1504.0 | 1510.5 | 1511.1 | 1511.0 | 1512.7 | 1507.7 | 1516.2 | 1503.6 | 1505.8 | 1493.4 |
| 0.78 | 1504.0 | 1510.5 | 1511.1 | 1511.0 | 1512.7 | 1507.7 | 1516.2 | 1503.6 | 1505.8 | 1493.4 |
| 0.79 | 1504.0 | 1510.5 | 1511.1 | 1511.0 | 1512.7 | 1507.7 | 1516.2 | 1503.6 | 1505.8 | 1493.4 |

## D.4 Validation of the Selected Parameters on POPE

Similarly, we conduct large-scale grid search experiments on all three POPE (MSCOCO)subsets: *Random*, *Popular*, and *Adversarial*. As shown in Tables 10, 11, and 12, the model performance remains stable across a wide range of $\tau$ and $\alpha$ settings. Specifically, optimal performance consistently emerges around $\tau = 0.74$ and $\alpha = 0.82$ for all subsets, indicating that our chosen configuration generalizes well across diverse hallucination scenarios in POPE. This consistent pattern further supports the effectiveness and reliability of our parameter selection strategy.

Table 10: Adversarial Accuracy across different $\tau$ and $\alpha$ values on LLaVA1.5-7b.

| $\tau\backslash\alpha$ | 0.81 | 0.82 | 0.83 | 0.84 | 0.85 | 0.86 | 0.87 | 0.88 | 0.89 | 0.90 |
|---|---|---|---|---|---|---|---|---|---|---|
| 0.72 | 0.805 | 0.805 | 0.805 | 0.803 | 0.803 | 0.802 | 0.802 | 0.802 | 0.801 | 0.800 |
| 0.73 | 0.807 | 0.807 | 0.807 | 0.806 | 0.805 | 0.804 | 0.803 | 0.802 | 0.801 | 0.800 |
| 0.74 | 0.807 | **0.807** | 0.807 | 0.806 | 0.805 | 0.804 | 0.803 | 0.802 | 0.801 | 0.800 |
| 0.75 | 0.807 | 0.807 | 0.807 | 0.806 | 0.805 | 0.804 | 0.803 | 0.802 | 0.801 | 0.800 |
| 0.76 | 0.807 | 0.807 | 0.807 | 0.806 | 0.805 | 0.804 | 0.803 | 0.802 | 0.801 | 0.800 |

Table 11: Popular Accuracy across different $\tau$ and $\alpha$ values on LLaVA1.5-7b.

| $\tau\backslash\alpha$ | 0.81 | 0.82 | 0.83 | 0.84 | 0.85 | 0.86 | 0.87 | 0.88 | 0.89 | 0.90 |
|---|---|---|---|---|---|---|---|---|---|---|
| 0.72 | 0.866 | 0.866 | 0.866 | 0.865 | 0.865 | 0.865 | 0.865 | 0.863 | 0.863 | 0.863 |
| 0.73 | 0.868 | 0.867 | 0.867 | 0.867 | 0.865 | 0.865 | 0.865 | 0.863 | 0.863 | 0.862 |
| 0.74 | 0.868 | **0.867** | 0.867 | 0.867 | 0.865 | 0.865 | 0.865 | 0.863 | 0.863 | 0.862 |
| 0.75 | 0.868 | 0.867 | 0.867 | 0.867 | 0.865 | 0.865 | 0.865 | 0.863 | 0.863 | 0.862 |
| 0.76 | 0.868 | 0.867 | 0.867 | 0.867 | 0.865 | 0.865 | 0.865 | 0.863 | 0.863 | 0.862 |

Table 12: Random Accuracy across different $\tau$ and $\alpha$ values on LLaVA1.5-7b.

| $\tau\backslash\alpha$ | 0.81 | 0.82 | 0.83 | 0.84 | 0.85 | 0.86 | 0.87 | 0.88 | 0.89 | 0.90 |
|---|---|---|---|---|---|---|---|---|---|---|
| 0.72 | 0.900 | 0.899 | 0.899 | 0.900 | 0.899 | 0.898 | 0.897 | 0.898 | 0.897 | 0.897 |
| 0.73 | 0.900 | 0.900 | 0.900 | 0.900 | 0.899 | 0.898 | 0.898 | 0.898 | 0.897 | 0.897 |
| 0.74 | 0.900 | **0.900** | 0.900 | 0.900 | 0.899 | 0.898 | 0.898 | 0.898 | 0.897 | 0.897 |
| 0.75 | 0.900 | 0.900 | 0.900 | 0.900 | 0.899 | 0.898 | 0.898 | 0.898 | 0.897 | 0.897 |
| 0.76 | 0.900 | 0.900 | 0.900 | 0.900 | 0.899 | 0.898 | 0.898 | 0.898 | 0.897 | 0.897 |

