# OpenReview forum: "Mitigating Hallucinations via Inter-Layer Consistency Aggregation in Large Vision-Language Models"
_ICLR.cc/2026/Conference — ICLR 2026 Conference Withdrawn Submission_

### Official Review · Reviewer_jLwM · 2025-10-26

**Soundness:** 3
**Presentation:** 3
**Contribution:** 2
**Rating:** 4
**Confidence:** 4

**Summary:**

The paper proposes Decoding with Inter-layer Consistency via Layer Aggregation (DCLA), a training-free method that mitigates hallucinations in language-vision models by enforcing semantic consistency across transformer layers during decoding. Extensive experiments demonstrate that DCLA effectively reduces hallucinations across MME and POPE.

**Strengths:**

* The approach is simple yet novel, presenting a training-free method for hallucination mitigation that avoids retraining or fine-tuning, making it both practical and widely applicable across different models and settings.

* The paper is well written and clearly structured, with explanations that make the methodology and results easy to understand and follow.

**Weaknesses:**

* __Hyperparameter selection__: It is unclear how the hyperparameters for the proposed method were chosen. If they were optimized using the test set, this could compromise the validity of the results. Clarification on whether a separate training or validation set was used is needed.

* __Missing benchmark evaluations__: The paper does not report results on established hallucination benchmarks such as CHAIR [1] or AMBER [2], which would provide a more comprehensive evaluation of the method’s effectiveness.

* __Limited model coverage__: Most results are presented on LLAVA-type models. Including experiments on stronger base models (e.g., Qwen2.5-VL or Qwen3-VL) would strengthen the paper by demonstrating broader applicability and robustness.

[1] Rohrbach et al., Object Hallucination in Image Captioning, EMNLP 2018

[2] Wang et al., AMBER: An LLM-Free Multi-Dimensional Benchmark for MLLMs Hallucination Evaluation, arXiv 2023

**Questions:**

It would be interesting to explore whether the proposed approach can be combined with other sampling strategies, such as nucleus sampling or beam search, which are known to influence hallucination behavior. Could such combinations further improve the results or provide additional robustness in mitigating hallucinations?

---

### Official Review · Reviewer_1gmE · 2025-10-30

**Soundness:** 1
**Presentation:** 2
**Contribution:** 2
**Rating:** 2
**Confidence:** 5

**Summary:**

This paper proposes an inter-layer consistency method that constructs a semantic reference by aggregating the previous layers and then corrects the deeper layer through a linear combination.

**Strengths:**

1. The experiments are extensive.

**Weaknesses:**

1. The performance gain on POPE (Table 3) is **marginal** across all settings (improvements within 1%). It’s unclear whether these gains exceed the standard deviation.

2. A major concern is that some claims are not well supported by experiments or prior studies, which makes the argument confusing. For example, in line 229, “However, this unidirectional information flow can lead to the gradual loss of fine-grained semantic cues captured by earlier layers.” This claim lacks empirical evidence or citation.

**Clearification**
1. Please fix the bolding in Tables 2 and 3. Both the regular baseline (or DAMO baseline) and your method achieve the same results, *why only your method is bolded*.

2. In Figure 3(a), the legend appears inconsistent with the plotted lines.

**Questions:**

1. The results in Table 1 are not convincing enough to support the conclusion in line 246: “These observations suggest that correction should not be applied indiscriminately but rather selectively based on the intrinsic characteristics of each layer.” In general, I don’t think Table 1 provides sufficient motivation for the proposed dynamic layer correction in Section 3.4.

2. Could you extend your experiments to include image description tasks (e.g., CHAIR evaluation) beyond yes/no tasks?

---

### Official Review · Reviewer_nMTh · 2025-11-01

**Soundness:** 3
**Presentation:** 3
**Contribution:** 2
**Rating:** 4
**Confidence:** 5

**Summary:**

This paper addresses the critical issue of hallucinations in Large Vision-Language Models (LVLMs) by proposing a novel, training-free decoding mechanism called Decoding with Inter-layer Consistency via Layer Aggregation (DCLA). The core motivation stems from the observation that hallucinations originate from localized surges in later layers, which override the more robust, visually-grounded semantic information encoded in earlier layers. DCLA aggregates information from these early, stable layers to refine the unstable predictions from deeper layers. The authors validate the approach on dedicated hallucination benchmarks (MME, POPE) and general VQA datasets (VizWiz, MM-Vet).

**Strengths:**

1. Well-Motivated Method: The method's design is highly motivated and clearly articulated, based on a compelling analysis of LVLM internal dynamics. The premise that: (1) "hallucinations often manifest as localized surges at the later layers, which tend to override earlier, visually grounded information," and (2) "Earlier layers encode basic semantic structures that remain robust to overfitting and noise introduced in deeper layers," provides a strong foundation for the proposed DCLA framework.

2. Sufficient Experiments: The paper conducts a comprehensive set of experiments. Evaluation covers both specialized metrics for hallucination mitigation (on MME and POPE datasets) and general VQA performance on established benchmarks (VizWiz and MM-Vet), providing sufficient empirical evidence to the performance verification of the proposed DCLA.

**Weaknesses:**

Major Concerns

While the foundational idea of leveraging stable semantic references from early layers is novel, the implementation of the DCLA mechanism, specifically the layer aggregation method (especially the weight calculation) and the strategy for selecting which layers to correct, appears to be heavily based on heuristics, necessitating a very strong empirical validation (significant performance gains).

That is to say, given the heuristic design, the practical efficacy is paramount. However, the performance improvements demonstrated on the critical hallucination mitigation benchmarks (MME, POPE) are limited, often hovering around or less than a 1% increase. Furthermore, some cases show performance consistent with the regular baseline. This limited gain raises concerns about the significance and robustness of the method.

Minor Concern

Formula (4) for the weight calculation does not explicitly contain the  $\gamma$, yet the text mentions a variable $\gamma$ controlling the decay based on layer distance. Clarification is needed on whether this hyperparameter is actually utilized in the final implementation and, if so, how it is incorporated into the mathematical formulation.

**Questions:**

1. Analysis of Drastic Performance Drop in Early Layers (Figure 3(a)): Figure 3(a) shows a particularly sharp performance drop with correction introduced on layers 0-32. Can the authors provide more explanations for this phenomenon? Furthermore, can this insight be leveraged to propose a more refined, or perhaps more computationally efficient, layer aggregation and decoding strategy?

2. Discrepancy in Model Effectiveness: According to Figure 3(b), the proposed DCLA method demonstrates a noticeably more significant effect on hallucination mitigation for the LLaVA1.5-7B model compared to the other three models tested. Can the authors explain the underlying reason for this model-specific discrepancy? Does the prominent performance gain on LLaVA1.5-7B, relative to its larger architectural siblings (LLaVA1.5-13B, LLaVA1.5-NEXT), suggest that DCLA's mechanism is inherently more effective at mitigating the specific type of hallucinations present in models with a lower baseline performance?

---

### Official Review · Reviewer_c4Xd · 2025-11-04

**Soundness:** 2
**Presentation:** 2
**Contribution:** 2
**Rating:** 2
**Confidence:** 4

**Summary:**

The paper proposes DCLA as a training-free decoding strategy for mitigating hallucinations in LVLMs. The method aims to enforce semantic consistency across transformer layers by aggregating hidden representations from earlier layers into dynamic reference and adaptively correct later layers when semantic drift is detected. Evaluations are done on models including LLaVA-v1.5 (7B/13B), LLaVA-NEXT, and mPLUG-Owl2. Benchmarks include MME, POPE, VizWiz and MM-Vet. Authors showed better performance as compared to some decoding-based baselines (DoLa, DAMO, VCD).

**Strengths:**

- Method is training-free, i.e. can be integrated into existing LVLMs without retraining.
- The paper evaluates across multiple LVLMs and benchmarks, showing improvements.
- The dynamic layer correction mechanism is interesting.

**Weaknesses:**

- The mechanism for adaptive correction (triggering threshold based on cosine similarity) is heuristic. No justification is provided for the choice of metric, threshold, or weighting function. In ablation, it's further observed that model is sensitive to the hyperparameter and table 6 shows very different number for different models. The optimal values for α = 0.82 and τ = 0.74 is too specific without any intuitive guidance. It seems impractical to tune hyperparameters for this method.
- The experiments heavily focused on LLaVA architecture (mPlug-owl2 is very similar). It's unclear how the method generalizes to other vl models including gemma3 and qwen-vl series.
- Performance seems on-par with the baseline DAMO.
- Missing latency or computational cost analysis. It's unclear how much latency the method induces during inference time.
- Inadequate discussion on the many previous works on visual hallucination reduction. [1-6]

[1] Opera: Alleviating hallucination in multi-modal large language models via over-trust penalty and retrospection-allocation.

[2] Mitigating Object Hallucination in Large Vision-Language Models via Image-Grounded Guidance.

[3] HALC: Object Hallucination Reduction via Adaptive Focal-Contrast Decoding.

[4] Don't Miss the Forest for the Trees: Attentional Vision Calibration for Large Vision Language Models.

[5] Reducing Hallucinations in Vision-Language Models via Latent Space Steering.

[6] Contrastive Region Guidance: Improving Grounding in Vision-Language Models without Training.

**Questions:**

N/A

---

### Note · Authors · 2026-01-04

I have read and agree with the venue's withdrawal policy on behalf of myself and my co-authors.